# Gradual Structured Pruning for Efficient Network Scaling in Image-Based Deep Reinforcement Learning

## Abstract

Scaling neural networks in image-based deep reinforcement learning often fails to improve performance. While it was shown that *unstructured* pruning of scaled networks can unlock performance gains, we find that refining the architecture of the scaled network yields even greater improvements. However, scaled networks in deep reinforcement learning present a practical challenge: the increased computational demands can hinder deployment on embedded devices, as commonly encountered in robotics applications. To address this, we propose a novel gradual group-structured pruning framework that allows performance gains through scaling while maintaining computational efficiency. Our method preserves the network's functional integrity of inter-layer dependencies in groups, such as residual connections, while seamlessly integrating with standard deep reinforcement learning algorithms. Experiments with PPO and DQN show that our approach sustains performance while significantly reducing inference time, making it the preferred approach for resource-limited deployment.

## 1 Introduction

Recent works on deep reinforcement learning (DRL) have revealed that apart from algorithmic improvements, considerable performance increases can come from the network architecture and training approach of the used deep neural networks (DNNs) themselves. Notably, Cobbe et al. (2020); Schwarzer et al. (2023); Obando-Ceron et al. (2024a) have shown that the Impala-CNN model (Espeholt et al., 2018), a 15-layer ResNet, outperforms the widely used convolutional neural network (CNN) model from Mnih et al. (2015) substantially. However, raising the parameter count of DNNs in DRL does not necessarily lead to improved performance (Schwarzer et al., 2023), as opposed to other areas in deep learning. Obando-Ceron et al. (2024a) provide a new perspective on scaling[1] DNNs in DRL by using unstructured magnitude pruning to increase sparsity gradually during training, which leads to a performance boost for Q-network-based DRL in Atari games.

Network pruning is a widely used technique in other deep learning fields , e.g., image classification (Vadera & Ameen, 2022), originally aimed at reducing DNNs' memory footprint and inference time but also known to frequently enhance robustness and generalization (Bartoldson et al., 2020). Its use in DRL may introduce advantageous regularization (Obando-Ceron et al., 2024a) but poses a unique challenge due to its dynamic training, requiring methods that maintain training stability over time. *Unstructured* pruning zeros out individual weight entries without considering their structural arrangements, such as filters and channels. This is in contrast to *structured* pruning, where such structures are entirely removed, directly reducing computational operations (Luo et al., 2017; He & Xiao, 2023) but leading potentially to high training instability.

We show in the following preliminary experiment that the benefits of network scaling in image-based DRL can also be unlocked by simple architectural refinements of the Impala-CNN, rendering the use of unstructured pruning from Obando-Ceron et al. (2024a) for performance increase obsolete. However, it opens the question of leveraging pruning in image-based DRL for the original motivation of lowering computational requirements, which is of high practical appeal for scaled DNNs.

---

[1]Scaling the width by increasing the number output channels per Conv2D layer by a factor $\tau$.

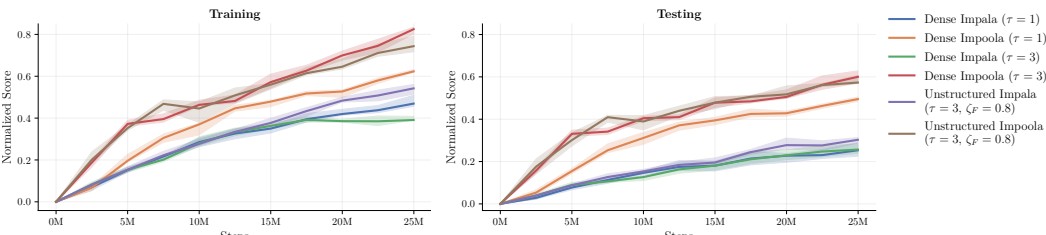

Figure 1: Effect of scaling the Impala and Impoola-CNN model's width from $\tau = 1$ to $\tau = 3$, evaluated on a subset of four ProcGen environments using PPO. Normalized return scores during training are shown for training levels (**left**) and testing levels (**right**). Gradual unstructured magnitude pruning (Obando-Ceron et al., 2024a) results are displayed for final target pruning rates $\zeta_F$ of 0.8. Appendix B.1 contains further results.

**Preliminary Experiment on Network Scaling:** We base this preliminary experiment on the Procgen benchmark (Cobbe et al., 2020), which is considered to challenge generalization better than Atari games. Figure 1 illustrates that scaling the width of the original Impala-CNN architecture to $\tau = 3$ does not result in performance improvements. Consistent with the findings of Obando-Ceron et al. (2024a), we observe that unstructured pruning of scaled Impala-CNNs enhances performance also for proximal policy optimization (PPO) within the Procgen environment. However, we demonstrate that scaling gains can also be realized by simply adding a *Pooling* layer before the Flatten layer of the Impala-CNN in combination with learning rate scheduling and weight decay–we name this architecture Im*pool*a-CNN. Note that classical ResNet models (He et al., 2016) also have this Pooling layer; the ablation in Figure B.4 shows that using one is crucial. The Impoola-CNN achieves significantly greater improvements than the use of unstructured pruning for scaled Impala-CNNs. Most notably, when using the scaled Impoola-CNN, the benefits of unstructured pruning vanish, even decreasing performance in training levels. Further results in Appendix B exhibit similar trends for deep Q-networks (DQNs) and include an additional supervised learning example. Thus, we encourage using the Impoola-CNN model for image-based DRL as it unlocks performance gains by network scaling directly, *without* the need for unstructured pruning.

**Structured Pruning for Efficient Scaling:** However, the use of scaled network architectures renders new practical problems in the form of increased memory footprint and computational requirements. This has particular implications for many DRL applications, e.g., robotics (Funk et al., 2022) or autonomous driving (Trumpp et al., 2023), as such applications are eventually deployed to resource-limited embedded devices with high control frequency requirements. This situation brings us back to the original notion of pruning to *reduce* computation requirements. As unstructured pruning only sets weights to zero, it often does not translate to a reduction in real-world inference times (Cheng et al., 2024). *Structured* pruning can be seen as a remedy since complete structures are removed from the DNN, thus reducing run times straight away (Luo et al., 2017). The feasibility of structured pruning in image-based DRL has been unexplored yet despite its practical appeal.

This paper establishes a framework for gradual group-*structured* pruning in image-based DRL, designed to reduce the computational requirements of scaled DNNs while closely matching the performance of dense baselines. To this end, we center our work on the Impoola-CNN since it not only outperforms the Impala architecture but also achieves greater efficiency with a reduced parameter count. Our study is based on the Procgen Benchmark (Cobbe et al., 2020) as this is the ideal evaluation platform to assess generalization, but we provide a supplementary experiment for Atari games. We discuss practical aspects such as fine-tuning capabilities and noise robustness and measure single-sample inference time, an aspect often overlooked by other works. Our main analysis uses PPO as this is the common baseline algorithm for the Procgen Benchmark but we provide additional results for DQN to cover a Q-network-based method.

Our main contributions are the following:

- We identify architectural limitations in the original Impala-CNN and propose the improved Im*pool*a-CNN model that unlocks performance gains through network scaling.
- Our gradual group-structured pruning framework accounts for inter-layer dependencies and enables performance gains through scaling while maintaining computational efficiency.

- We provide extensive experiments comparing various pruning methods, including measured inference times on diverse platforms. Our analysis makes a strong practical case for our group-structured method as it meets the performance of other methods while lowering compute demand substantially.

- The used source code will be made publicly available.

## 2 RELATED WORK

**Neural Network Pruning:** Neural network pruning is a technique to reduce compute time and/or memory size of a DNN by removing its weights, ideally without a substantial loss of accuracy (Han et al., 2016). Networks can be pruned in single or multiple steps with subsequent fine-tuning after training or gradually over its course (Cheng et al., 2024). Structured pruning (He & Xiao, 2023) leads to universal speed-ups as complete structures, e.g., filters or neurons, are removed. As single weights are set to zero in unstructured pruning, inference speed is not necessarily reduced but only theoretical FLOPS (Luo et al., 2017). Specialized hard- and software, which may improve the computation of such sparse kernels, e.g., semi-structured (2:4) patterns (Mishra et al., 2021), makes it slowly into the mainstream. Modern network architectures, e.g., ResNets of Transformers, pose complex structural dependencies that must be captured for correct pruning (Fang et al., 2023).

**Sparsity in Deep Reinforcement Learning:** Compared to computer vision (CV), exploring sparsity and pruning in DRL is a relatively recent effort. Livne & Cohen (2020) demonstrated that it is feasible to sparsify DRL agents during training without performance degradation. Various methods that sparsify agents during training are discussed by Yu et al. (2020); Tan et al. (2023); Sokar et al. (2021); Su et al. (2024). However, it was revealed by Graesser et al. (2022) that magnitude pruning during training of DRL agent with a gradually increasing target sparsity outperforms such methods.

**Scaling in Deep Reinforcement Learning:** The subsequent work of Obando-Ceron et al. (2024a) further investigates the effect of gradual unstructured pruning. When pruning a scaled Impala-CNN, they discover magnified performance in Atari for Q-network-based DRL. However, performance stays mostly the same for soft actor-critic (SAC) and PPO with dense networks in Mujoco, assumably because Mujoco is not image-based. It was shown that training can also be stabilized by using mixtures of experts to scale the Dense layer of the Impala-CNN (Obando-Ceron et al., 2024b), or by incorporating auxiliary tasks during training (Farebrother et al., 2023).

**Generalization in Deep Reinforcement Learning:** Zhang et al. (2018) reveal that DRL agents can memorize a non-trivial number of training levels, even with completely random rewards. Similar experiments in (Cobbe et al., 2019) quantify that the use of the same environment for both training and testing results in high overfitting of DRL agents. They show that well-known techniques from supervised learning, e.g., $L_2$ regularization, batch normalization, and data augmentation, reduce overfitting. However, only slightly better test performance is achieved when combining them than using them individually. Overfitting in DRL may be associated with a loss of network plasticity (Nikishin et al., 2022; Sokar et al., 2023). (Cobbe et al., 2020) introduces the Procgen Benchmark with various procedurally generated environments to measure sample efficiency and generalization.

## 3 BACKGROUND

### 3.1 DEEP REINFORCEMENT LEARNING

The iterative optimization in model-free DRL is formalized by a Markov decision process (MDP) with tuple $(\mathcal{S}, \mathcal{A}, \mathcal{T}, \mathcal{R}, \gamma)$. Here, $\mathcal{S}$ and $\mathcal{A}$ represent the state and action spaces, respectively, while the transition function $\mathcal{T} : \mathcal{S} \times \mathcal{A} \to \mathcal{P}(\mathcal{S})$ defines the probability distribution over the next state given the current state and action. The reward function is defined as $\mathcal{R} : \mathcal{S} \times \mathcal{A} \to \mathcal{R}$ and $\gamma$ is a discount factor. The mapping $\pi : \mathcal{S} \to \mathcal{P}(\mathcal{A})$ is called a (stochastic) action policy. A DNN with weights $\theta$ parameterizes the policy $\pi_\theta$ in DRL. The optimal policy $\pi_\theta^*$ maximizes the expected return $V_{\pi_\theta}(s) = \mathbb{E}_{\pi_\theta} \left[ \sum_{t=0}^{\infty} \gamma^t \mathcal{R}(s_t, a_t) \mid s_0 = s_t \right]$.

**Q-Network Methods:** These DRL methods are typically based on an estimate of the q-value function $Q_\pi(s, a) := \mathbb{E}_{\pi_\theta} \left[ \sum_{t=0}^{\infty} \gamma^t \mathcal{R}(s_t, a_t) \mid s_0 = s_t, a_0 = a \right]$. This function can be learned iteratively by temporal difference learning (Sutton, 1988) and bootstrapping the current q-value esti-

mate. DQN (Mnih et al., 2015) implements this by training a DNN with loss function $L(\theta) = \mathbb{E}_{(s,a,r,s')\sim\mathcal{D}}\left[(r + \gamma \max_{a'} Q(s', a'; \theta^-) - Q(s, a; \theta))^2\right]$ where transitions $(s, a, r, s') \sim \mathcal{D}$ are sampled from the experience replay buffer $\mathcal{D}$ and by using a target network with $\theta^-$ as delayed copies of $\theta$. Actions are obtained greedily by $a^* = \arg\max_a Q(s, a; \theta)$. The performance of vanilla DQN can be vastly improved by incorporating techniques such as double q-learning (Van Hasselt et al., 2016), multi-step rewards (Sutton, 1988), prioritized replay buffer (Schaul et al., 2015), and distributional q-learning (Bellemare et al., 2017), eventually forming Rainbow (Hessel et al., 2018).

**Actor-Critic Methods:** In addition to a critic network, e.g., $V(s; \phi)$ that estimates the state value, the action policy is defined as a dedicated actor network that can be directly optimized towards an optimization goal. PPO (Schulman et al., 2017) is an *on*-policy DRL method, where the weights $\theta$ are updated with respect to the advantage function $A(s, a) = Q(s, a) - V(s)$. The generalized advantage estimate (GAE) (Schulman et al., 2015) is the common choice to estimate $A(s, a)$. The loss (clip version) of the PPO actor for a transition tuple $e = (s, a, r, s')$ of a trajectory $\tau = \{e, e', ...\}$ is given by $L(\theta) = \mathbb{E}_t\left[\min\left(r(\theta)A, \text{clip}(r(\theta), 1 - \epsilon, 1 + \epsilon)A\right)\right]$. Here, $r(\theta) = \frac{\pi_\theta(a|s)}{\pi_{\theta_{\text{old}}}(a|s)}$ is the probability ratio between the old and new policy, where the hyperparameter $\epsilon$ limits their deviation.

**Impala-CNN:** The Impala-CNN was introduced by Espeholt et al. (2018) as a 15-layer ResNet model for encoding image inputs. The architecture combines two building blocks. ConvSequence $S_j$ blocks consist first of a Conv2D layer with MaxPooling and ReLU activation and then 2 subsequent ResBlock blocks as $S_j : \{C_j \to P \to R_{0,j} \to R_{1,j}\}$; the ResBlock blocks are based on two Conv2D layers with ReLU activation and a residual connection $R_{i,j} : \{C_{0,i,j} \to C_{1,i,j}\}$. The vanilla Impala-CNN stacks three ConvSequence blocks $\{S_0, S_1, S_2\}$ with each block having the same amount of convolutional output channels $\{c_0^{\text{out}}, c_1^{\text{out}}, c_2^{\text{out}}\} = \{16, 32, 32\}$; scaled network versions multiply this configuration by a width scaling factor $\tau$. The original implementation by Espeholt et al. (2018) uses a Linear layer of 256 neurons as the last encoder layer.

## 3.2 NEURAL NETWORK PRUNING

Assume an initial DNN $f_\theta$ with parameters $\theta = \{w_1, w_2, \dots\}$ of a parameter space $\mathcal{H}$ and weight tensors $w \in \mathcal{R}^{N \times M \times \cdots}$. Let $c(f)$ be a counting function that counts the number of parameters in a DNN. We then define an arbitrary pruning operation as a function

$$p : f_\theta \to f_{\theta'} \text{ with } c(f_\theta) \geq c(f_{\theta'}) \text{ and } \theta' \subseteq \theta; \tag{1}$$

This operation leads to the pruned network $f_{\theta'}$ with parameters $\theta'$ and sparsity $\zeta = 1 - \frac{c(f_{\theta'})}{c(f_\theta)}$.

**Importance Score:** The selection of which parameters to prune is based on a score function that estimates the importance of each parameter $\iota : \mathcal{H} \to \mathcal{R}$. This score defines the order of parameters to be pruned, i.e., $w_{1,1}$ will be pruned first when $\iota(w_{1,2}) > \iota(w_{1,1})$. Various criteria are discussed in the literature (Cheng et al., 2024), e.g., weight magnitude, saliency, and Taylor expansions; random weights selection often meets their performance (Li et al., 2022; Liu et al., 2022).

**Unstructured Pruning:** Specific entries in $w$ are set to zero, but the overall tensor shape is kept. $\dim(w') = \dim(w)$. Inference time is not necessarily reduced.

**Structured Pruning:** Full structures of the weight matrix $w$ are removed, reducing the size of the weight matrix. For example, pruning of a single output channel of a Conv2D layer with a weight tensor shape[2] of $\{48, 32, 3, 3\}$ leads to a new tensor with shape $\{47, 32, 3, 3\}$.

**Gradual Pruning:** Gradual pruning in DRL (Graesser et al., 2022) involves progressively removing parameters from the network throughout the training process by applying a series of pruning steps $p_F(\dots(p_2(p_1(f))))$. Typically, this process follows a predefined schedule with a target sparsity $\zeta_{t,T}$ at each pruning step $t$ to define the number of parameters to be pruned. The pruning schedule begins after a warmup training phase at step $t_{\text{start}}$ and concludes with the final pruning operation at step $t_{\text{end}}$. The final target sparsity $\zeta_F$ represents the fraction of the remaining parameters. A commonly used gradual pruning schedule is a third-order polynomial with $t_{\text{start}}$ and $t_{\text{end}}$ set to 20% and 80% of the total training steps, respectively (Graesser et al., 2022; Obando-Ceron et al., 2024a).

---

[2]The weight matrix of a Conv2D layer has dimension $\{C_{\text{out}}, C_{\text{in}}, K, K\}$ with the number of out and in channels $C_{\text{out}}$ and $C_{\text{out}}$, respectively, and a kernel of shape $K \times K$.

Figure 2: Visualization of the effect of pruning techniques on the Impoola/Impala-CNN's ResNet architecture. Due to the residual connection, there is a dependency between Conv2D layers. Here, the output channels (**blue**) of the last Con2D layer must have the same dimension as the output channels of the Con2D layer (**red**) before the ResBlock. Unstructured and naive structured pruning does not account for this when pruning the first Con2D layer (**red**). Only group-structured pruning removes the same channels and corresponding filters correctly, including the following layer's unnecessary filters (**gray**).

### 3.3 DEPENDENCY GRAPH

Weight tensors $w$ in DNNs exhibit inter-layer dependencies, e.g., the output dimension of one layer defines the input dimension of the next. Consequently, pruning parameters in one layer may necessitate further pruning in the dependent layers. These group-structured parameters form a graph that models the dependencies between network layers (Fang et al., 2023). While building such a dependency graph for networks composed solely of linear layers is straightforward, modern architectures with residual connections or attention layers introduce additional complexity, requiring automated methods. Fang et al. (2023) propose DepGraph, a generic framework that uses graph traversal to identify the dependency graph $D$ and its dependencies. A parameter group $g = \{w_1, w_2, \cdot\}$ is a subgraph of $D$ and must be pruned simultaneously to maintain the network's functional integrity.

## 4 METHODOLOGY

As outlined in Section 1, our preliminary experiments reveal that our proposed Impoola-CNN model leads to an overall performance boost, but gains attributed to unstructured pruning vanish. Thus, we introduce an approach using *structured* pruning instead with the motivation to reduce compute time but without degrading performance. Our gradual *group-structured* pruning framework can be plugged into existing DRL algorithms easily; we show this for PPO and DQN agents. Our method accounts for dependencies of the Impoola/Impala-CNN encoders, which is crucial for performance.

### 4.1 IMPOOLA-CNN

In contrast to the Impala-CNN (Espeholt et al., 2018), the Impoola-CNN simply adds an Average-Pooling layer after the last Con2D layer as listed in Table E.5. The overview in Appendix E shows that for the Impala model, 64.19 % of the weights are located in the encoder's last Linear layer, while weights in the Impoola-CNN are equally distributed over the network with 10.1 % in the last layer. We speculate that this balanced distribution, specifically reducing the number of Linear layer weights, contributes to the significant performance improvements of the Impoola-CNN.

### 4.2 GRADUAL GROUP-STRUCTURED PRUNING

**Dependency Graph:** We use a dependency graph (Fang et al., 2023) to correctly identify parameter groups $\{g_1, g_2, \dots\}$ in the DNN that should be pruned simultaneously. Figure 2 visualizes the need to correctly account for dependency introduced by the residual connection in the Impala/Impoola-CNN model. Opposed to our used *group-structured* pruning method, unstructured and naive structured pruning does not account for such dependencies, altering the nature of the DNN's residual connections. We define our pruning approach as the function $p(g, N)$ per group $g$ with $N$ as the number of structures to be pruned. The pruning operation $p$ process groups $g = \{w_1, w_2, \dots\}$ and assigns importance scores $\iota(w_1, w_2, \dots)$ for the common tensor dimension of the group.

**Group Importance Scoring:** Following other works (Graesser et al., 2022; Obando-Ceron et al., 2024a), we define the weight magnitude , i.e., L1-norm, as scoring function $\iota(w) = ||w||_1$ to be used along the common tensor dimension independently for each group. The final importance score of each structure in $g$ is obtained by normalizing $\iota(w)$ first for each structure's weight tensor. Next, we take the mean of these local scores in group $g$ as a reduction function to obtain the final scores, which creates an aggregated score vector for the group's common tensor dimension. This reduction means that our method takes a neuron's intra-layer dependencies into account for its score.

**Gradual Pruning Schedule:** We utilize the same third-order polynomial pruning scheme with $t_{\text{start}} = 20\%$ and $t_{\text{end}} = 80\%$ as done in other works (Graesser et al., 2022; Obando-Ceron et al., 2024a). At each time step during training, the current sparsity of all layers $l_i$ is measured as $\zeta_{l_i}$. The number of structures to be pruned is then calculated as

$$N_i = \lfloor (\zeta_t - \zeta_{l_i}) \cdot ||w_{l_i}||_0^{\text{init}} \rfloor, \tag{2}$$

where $N_i$ represents the number of structures to be pruned, and $|w_{l_i}||_0^{\text{init}}$ is the initial number of structures in layer $l_i$. Opposed to unstructured pruning where single weight entries can be removed, using Equation 2 often results in no structures being pruned at certain steps due to the floor operator. However, when a structure is pruned, it leads to the removal of many parameters at once. For example, pruning an output channel of a Conv2D layer with 48 input channels and a 3x3 kernel results in the simultaneous removal of 432 parameters. Thus, unstructured pruning allows for more gradual and fine-grained reductions, as illustrated in Figure C.11 and C.13, respectively.

## 4.3 IMPLEMENTATION DETAILS

**Deep Reinforcement Learning Agents:** We use PPO and DQN agents in this work. Our implementations are derived from CleanRL (Huang et al., 2022) for PyTorch (Paszke et al., 2017). Hyperparameters are listed in Appendix D. The used DQN agent is extended by double q-learning (Van Hasselt et al., 2016), multi-step rewards (Sutton, 1988), and prioritized replay buffer (Schaul et al., 2015). We use the framework from Fang et al. (2023) to derive the dependency graph $D$, allowing us to deploy the correct structured pruning of inter-layer dependencies. The unstructured and naive structured pruning methods use weight masks from PyTorch (Paszke et al., 2017).

**Network Architecture:** We deploy the Impala/Impoola-CNN encoders with an output feature dimension of 256 in all experiments. We set $\tau = 3$ for all experiments unless otherwise specified, as suggested by other works (Obando-Ceron et al., 2024a). The CNN encoder is shared between the actor and critic for PPO. Given an image input of 64x64 pixels, the Impala and Impoola-CNN consists of 2,450,640 and 976,080 trainable parameters for PPO, respectively.

**Regularization:** Our Impoola-CNN model uses a weight decay of $1e^{-5}$. Linear learning rate annealing rate is used for the PPO agent, which greatly improves performance. It was shown by Li et al. (2019) that as learning rate annealing may allow for higher initial learning rates, generalization can be improved. We provide an ablation study on this in Section 5.4. No learning rate schedule is used for DQN as this reduced performance in environments with sparser reward; see Appendix B.2.

## 5 EXPERIMENTS

Unless otherwise stated, the results presented are based on the Impoola-CNN model, as we showed its superiority against the Impala-CNN. The experiments are conducted for a subset of four environments for the Procgen Benchmark (Cobbe et al., 2020); see Appendix A for their description. Our evaluation strongly focuses on measuring the generalization of DRL agents, for which Atari games are unsuitable. To keep compute requirements reasonable, results are based on the *easy* game setting with the configuration as recommended by Cobbe et al. (2020). The presented scores are median results and 95-% confidence intervals, using 5 seeds for each environment per experiment and 2,500 evaluation episodes. We report collected returns as normalized scores $S$ according to Equation A.1, where 1.0 corresponds to an optimal policy and 0.0 is equivalent to a random one.

**Evaluation Tracks:** We provide an extensive evaluation by introducing the following tracks:

1. *Generalization*: The agent is trained for 25M interaction steps on 200 training levels but then evaluated on the full distribution as testing levels, thus evaluating generalization.

Table 1: Average final normalized scores (return) to compare pruning methods for PPO with a scaled Impoola-CNN of scale $\tau = 3$. Total training times for a single NVIDIA A100 PCIe 40GB GPU. An [↑] indicates higher values mean better performance. We highlight each of the best dense and sparse results in bold font.

| $\zeta_F$ | Method | $S_{\text{Training}}$ [↑] | $S_{\text{Generalization}}$ [↑] | $S_{\text{Fine-tuning}}$ [↑] | $S^{\sigma=5\|\sigma=15}_{\text{Robustness}}$ [↑] | Training [↓] |
|---|---|---|---|---|---|---|
| - | Dense (w/o Impoola) | $0.39^{\pm 0.02}_{\pm 0.02}$ | $0.26^{\pm 0.02}_{\pm 0.02}$ | $0.37^{\pm 0.02}_{\pm 0.02}$ | $0.25^{\pm 0.03}_{\pm 0.02}$ \| $0.23^{\pm 0.01}_{\pm 0.02}$ | 3h:04 |
| - | Dense | $0.82^{\pm 0.03}_{\pm 0.02}$ | $0.60^{\pm 0.03}_{\pm 0.04}$ | $0.70^{\pm 0.04}_{\pm 0.02}$ | $\mathbf{0.60}^{\pm 0.03}_{\pm 0.04}$ \| $0.56^{\pm 0.01}_{\pm 0.03}$ | 2h:24 |
| - | ReDo | $\mathbf{0.85}^{\pm 0.02}_{\pm 0.01}$ | $\mathbf{0.63}^{\pm 0.02}_{\pm 0.02}$ | $\mathbf{0.72}^{\pm 0.02}_{\pm 0.02}$ | $0.59^{\pm 0.02}_{\pm 0.03}$ \| $\mathbf{0.59}^{\pm 0.02}_{\pm 0.03}$ | 3h:32 |
| 0.8 | Distillation BC | $0.73^{\pm 0.01}_{\pm 0.03}$ | $0.51^{\pm 0.0}_{\pm 0.01}$ | - | $0.50^{\pm 0.01}_{\pm 0.01}$ \| $0.50^{\pm 0.01}_{\pm 0.01}$ | 3h:59 |
| 0.8 | Unstructured | $\mathbf{0.74}^{\pm 0.04}_{\pm 0.06}$ | $\mathbf{0.57}^{\pm 0.02}_{\pm 0.02}$ | $\mathbf{0.61}^{\pm 0.02}_{\pm 0.06}$ | $\mathbf{0.57}^{\pm 0.02}_{\pm 0.02}$ \| $0.55^{\pm 0.01}_{\pm 0.01}$ | 2h:25 |
| 0.8 | Naive Structured | $0.59^{\pm 0.01}_{\pm 0.02}$ | $0.51^{\pm 0.01}_{\pm 0.01}$ | $0.54^{\pm 0.02}_{\pm 0.01}$ | $0.51^{\pm 0.02}_{\pm 0.01}$ \| $0.51^{\pm 0.02}_{\pm 0.02}$ | 2h:23 |
| 0.8 | Group-Structured | $0.72^{\pm 0.03}_{\pm 0.04}$ | $\mathbf{0.57}^{\pm 0.02}_{\pm 0.02}$ | $\mathbf{0.61}^{\pm 0.02}_{\pm 0.05}$ | $\mathbf{0.57}^{\pm 0.02}_{\pm 0.01}$ \| $\mathbf{0.57}^{\pm 0.01}_{\pm 0.01}$ | **1h:37** |

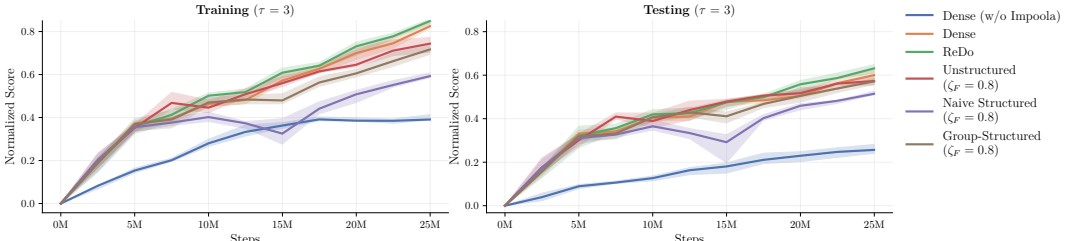

Figure 3: Normalized score (return) during training PPO using our Impoola-CNN. Evaluated on training (**left**) and testing levels (**right**) every 2.5M steps for 2,500 episodes.

2. *Fine-tuning*: A similar setting to generalization with 200 initial training levels. After training for 25M initial steps, the agent is fine-tuned for another 1M steps on 100 additional levels. Performance is then evaluated only on these 100 additional levels.

3. *Noise Robustness*: We follow Graesser et al. and use input perturbation with sampled noise $x \sim \mathcal{N}(0, \sigma)$ with $\sigma \in [5, 15]$ add to each pixel in the observation space as an integer.

**Baseline Methods:** We compare our *group-structured* pruning framework with unstructured pruning (Obando-Ceron et al., 2024a) and naive structured pruning, which does not account for inter-layer dependencies. Further results are given for a distillation method that uses behavior cloning (BC) with a dataset of 10M examples collected with the trained dense model to distill it into a reduced-size network, equivalent to the pruned networks. Additionally, we include results for ReDo (Sokar et al., 2023), which does not remove but re-invoke neurons that do not contribute to the model's output, so it can be interpreted as inverse pruning.

## 5.1 RESULTS FOR PPO

**Training and Generalization:** Our first experiment in Figure 3 evaluates our group-structured pruning method for PPO during training for the generalization track. Although both unstructured and group-structured pruning methods result in some performance loss on training levels, their generalization capabilities degrade only slightly compared to the dense baseline. The results on the training levels indicate that group-structured pruning is more invasive during the active pruning phase (from 5M to 20M steps) than unstructured pruning. However, it recovers the performance loss in the final 5M steps once pruning is completed. In contrast, the naive structured pruning approach exhibits overall degraded performance, stressing the importance of correctly handling dependencies, as achieved with our group-structured pruning method. Interestingly, Table 1 shows that the distillation BC method suffers from low generalization performance, proving the advantage or gradual pruning. The ReDo method exhibits the best final performance for both training and generalization. This challenges the efficacy of unstructured pruning, suggesting that reinitializing dormant neurons may be more beneficial than simply removing weights. Similarly, this finding also supports the case for *group-structured* pruning: while this method only causes a slight generalization degradation compared to Dense and ReDo, it offers the significant advantage of reduced training time as

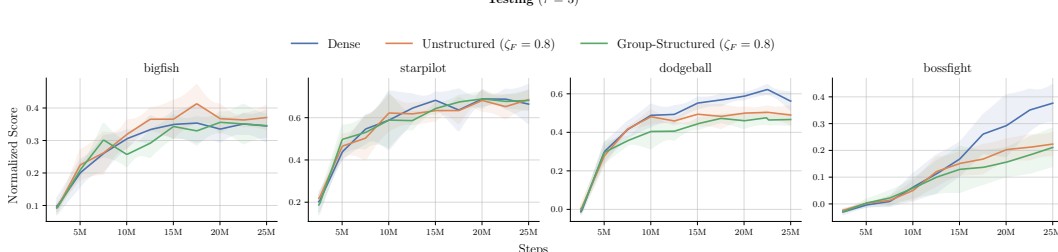

Figure 4: Normalized scores (return) per environment during DQN training, evaluated on test levels to measure generalization. The aggregated scores are visualized in Figure C.12.

Table 2: Latency times in ms across compute devices (see Appendix F) for the presented pruning algorithms using the Impoola-CNN model ($\tau = 3$). Batch size 1 mimics real-world inference applications of DRL.

| Device | Batch Size 256 | | | | | Batch Size 1 | | | | |
|---|---|---|---|---|---|---|---|---|---|---|
| | Dense | Unstructured | | Group-Structured | | Dense | Unstructured | | Group-Structured | |
| | | $\zeta_F$=0.8 | $\zeta_F$=0.9 | $\zeta_F$=0.8 | $\zeta_F$=0.9 | | $\zeta_F$=0.8 | $\zeta_F$=0.9 | $\zeta_F$=0.8 | $\zeta_F$=0.9 |
| High-end GPU | 11.4 | 11.4 | 11.4 | 6.3 | **4.1** | 1.0 | 1.0 | 1.0 | **0.8** | **0.8** |
| Workstation CPU | 337.4 | 342.7 | 342.7 | 144.7 | **63.3** | 3.0 | 3.2 | 3.2 | 2.6 | **1.4** |
| Embedded GPU | 383.9 | 383.9 | 383.9 | 266.5 | **165.3** | 6.8 | 6.8 | 6.8 | **6.3** | **6.3** |
| Embedded CPU | - | - | - | - | - | 32.5 | 27.6 | 27.4 | 14.0 | **9.6** |

the network gets gradually pruned during training. Additionally, it can be seen that our Impoola-CNN accelerates Dense training over the Impala-CNN, making a case for the combined use with group-structured pruning.

**Fine-tuning and Robustness:** Additional results for the fine-tuning and robustness tracks are presented in Table 1. Dense and ReDo methods achieve high fine-tuning scores, likely due to their strong initial generalization and access to a larger hypothesis space. This confirms that the architectural improvements of the Impoola-CNN are effective. Although pruned networks improve performance when fine-tuned on additional levels, their gains are more limited in comparison. However, the pruned networks demonstrate superior robustness under noisy observations, outperforming Dense and ReDo. This increased robustness may result from the reduced parameter count in pruned networks, which limits flexibility but may control internal activation.

## 5.2 RESULTS FOR DQN

We provide another study for DQN. As shown in Figure 4, we observe a similar trend to PPO, where the Dense Impoola-CNN constitutes the consistent performance across environments. Notably, group-structured pruning is slightly outperformed by unstructured pruning. This may be attributed to the higher frequency of gradient updates in DQN, which favors the smoother pruning schedule of unstructured pruning, where single weights are removed incrementally. In contrast, group-structured pruning involves fewer but larger pruning steps, resulting in a more step-like pruning scheme. Further analysis of Figure 4 reveals that the pruning methods underperform primarily in the Dodgeball and Bossfight environments. In the Bossfight environment, the DQN agent appears not to have learned a strong policy by the time pruning starts at 5M steps. We conclude that initiating pruning when the Dense agent's performance is still unstable can induce training instability, leading to further deterioration in performance. Appendix C contains the results on training levels for DQN.

## 5.3 INFERENCE TIMES

We present the measured inference times for the PPO actor using the Impoola-CNN model in Table 2. For a batch size of 256, our group-structured pruning method results in a significant reduction in inference times across all devices. This reduction is particularly beneficial for accelerating training or enabling on-board fine-tuning on embedded devices with batched training samples. In the evaluation for single-sample inference, it can be seen that the reduction on GPU platforms is less than for

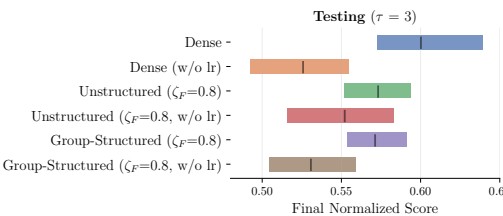

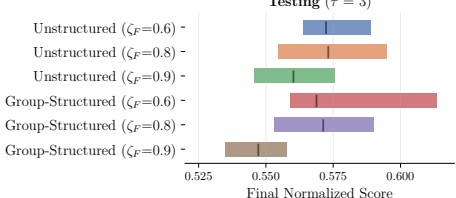

Figure 5: Ablation on Impoola-CNN without linear learning rate annealing (w/o lr).

Figure 6: Ablation on different final target pruning rates $\zeta_F$ using the Impoola-CNN.

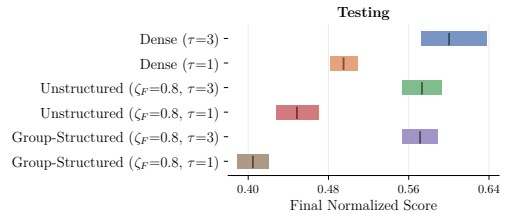

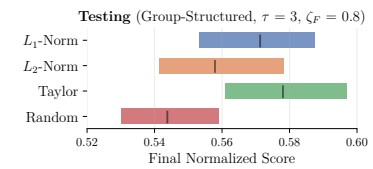

Figure 7: Ablation on reducing the width scale of the Impoola-CNN to $\tau = 1$.

Figure 8: Ablation on the importance score function $\iota$ for group-structured pruning with Impoola-CNN.

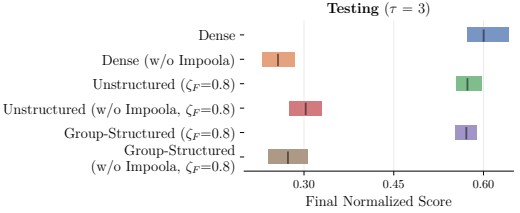

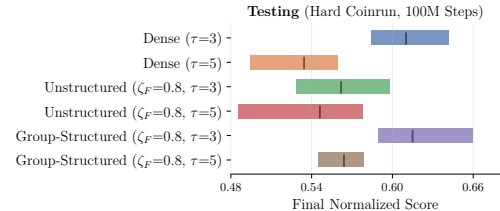

Figure 9: Ablation on using the Impala-CNN (= w/o Impoola) instead of the Impoola-CNN.

Figure 10: Additional results for the hard setting in Coinrun environment using 100M training steps.

CPU. This finding indicates that the used Impoola-CNN with a width scale $\tau = 3$ is under-utilizing the available GPU resources, limiting the potential gains. As we anticipate further growth in DNN sizes for image-based DRL, the results for batch size 256 highlight the considerable efficiency gains achievable with our group-structured approach when available computing power is fully utilized.

## 5.4 Ablations

**Learning Rate Annealing:** We examine the influence of linear learning rate annealing on the performance of PPO agents. Li et al. (2019) indicate that learning rate annealing improves generalization performance. However, this could interfere with pruning, as the network may adapt more slowly after pruning when using a lower learning rate. As shown in Figure 5, learning rate annealing plays a crucial role in generalization also for pruning methods with PPO. Notably, its use appears to reduce performance variance across the pruning methods.

**Pruning Ratio and Width Reduction:** Figure 6 visualizes performance for different target pruning rates $\zeta_F$. With a high target ratio $\zeta_F = 0.9$, both unstructured and group-structured pruning result in reduced performance. Group-structured pruning seems to be more sensitive to high pruning rates, positioning $\zeta_F = 0.8$ as a favorable compromise between maintaining performance and reducing computation time. As presented in Figure 7, reducing the width scale to $\tau = 1$ decreases overall performance, but group-structured pruning seems to be more sensitive than unstructured pruning, i.e., structured grouping appears to require a minimum amount of filters to work well.

**Importance Score:** We use the $L_1$-norm as importance score function $\iota$ to allow for easier comparison with other works (Obando-Ceron et al., 2024a). We ablate using instead $L_2$-norm, a first-order Taylor expansion of the loss (Molchanov et al., 2019), and random scoring in Figure 8. It can be seen that the Taylor-based score function may improve performance slightly. However, the overall influence of the importance score seems minor, as even random selection yields good performance.

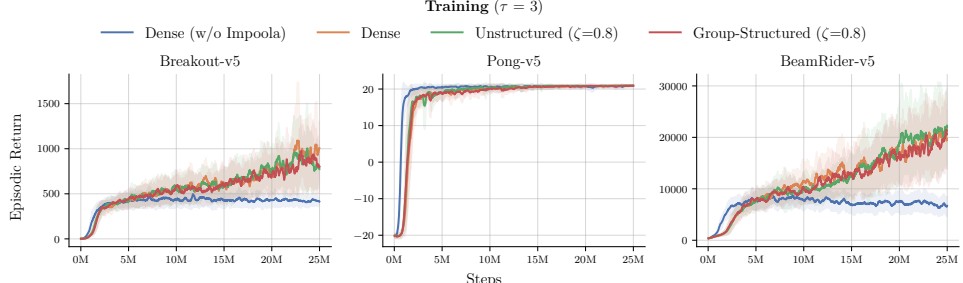

Figure 11: Episodic return in Atari games, parallelized in EnvPool (Weng et al., 2022), with training PPO for 25M steps using the same hyperparameter as for the Procgen Benchmark.

This finding aligns with recent works on random network pruning (Li et al., 2022; Liu et al., 2022). Overall, we recommend using $L_1$-norm due to its simplicity.

**Impala-CNN:** While this work focuses on the Impoola-CNN architecture, we present additional results for our group-structured method when using Impala-CNN. It can be seen in Figure 9 that performance for the Impala-CNN can be improved by using both unstructured and group-structured pruning, but the improvement is higher with unstructured pruning. However, as already discussed, Impoola-CNN is clearly the preferred architecture.

**Long-term Stability:** We investigate the effect of prolonged training through experiments on Coin-run with *hard* setting and 100M steps. Figure 10 shows that unstructured pruning for $\tau = 3$ outperforms the others slightly for training levels after an initial phase of instability. However, the results on testing levels reveal that this seems to come with some overfitting, as group-structured pruning is the best-performing method for generalization. Increasing the width scale to $\tau = 5$ reduces performance, though the degradation is least pronounced in group-structured pruning.

**Atari Games:** To demonstrate that Impoola-CNN and group-structured pruning are not specialized to the Procgen Benchmark, we run experiments on three Atari games: Breakout-v5, Pong-v5, and BeamRider-v5. The results in Figure 11 demonstrate first that the Impoola-CNN outperforms Impala-CNN significantly, establishing it as a generally applicable improvement. Second, group-structured pruning matches again the performance of the dense Impoola-CNN and unstructured pruning, confirming the results for Procgen Benchmark.

## 6 CONCLUSION AND FUTURE WORK

Following preliminary experiments, this work introduces the scaled Impoola-CNN encoder, which significantly boosts image-based DRL performance compared to the widely used Impala-CNN. We present a group-structured pruning framework for the Impoola-CNN that unlocks performance gains through scaling while maintaining computational efficiency. Our results on the Procgen Benchmark for PPO and DQN show that this pruning method for image-based DRL maintains performance comparable to that of networks with unstructured pruning, even outperforming dense DNNs for generalization in long-term training. Moreover, we show that while pruned networks do not adapt to fine-tuning levels as high as the dense Impoola-CNN, pruned DNNs show strong noise robustness instead. Additional results for Atari games demonstrate the broad applicability of our approach. A final analysis highlights the efficiency of our group-structured pruning method, with significant reductions in computing time, eventually making the case for group-structured pruning as the preferred approach for real-world image-based DRL applications with scaled networks.

For future work, evaluation for real-world image-based DRL applications could provide valuable insights, particularly given the demonstrated noise robustness of group-structured pruning. The fact that pruning methods seem to not outperform the Dense networks for the Impoola-CNN requires further analysis but can be related to the finding of (Cobbe et al., 2019) that if some regularization is already applied, the combination with further regularization technique does not necessarily improve performance further. Additionally, we see potential for exploring adaptive gradual pruning schedules and the use of global pruning methods.

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

# – Supplementary Material –

# Gradual Structured Pruning for Efficient Network Scaling in Image-Based Deep Reinforcement Learning

## A PROCGEN ENVIRONMENTS

The ProcGen environments were developed by Cobbe et al. (2020) to test sample efficiency and generalization of DRL agents. Unless otherwise stated, our results are based on the *easy* setting. For the *generalization* track, 200 levels are used for training, while all procedurally generated levels are used for evaluation. Our experiments have shown that the initial set of 200 levels can influence the agent's performance. Thus, we fix the level generation to the first 200 levels for all experiments but always report results for independent runs with different seeds for the training. When the *hard* setting is used, 1000 training levels are used. The action space of the ProcGen environments consists of 15 discrete actions. Observations are RGB images with 3x64x64 pixels. No stacking of images is required, as we utilize the environments without the setting that requires memory.

**Game Selection:** We chose Bigfish, Starpilot, Dodgeball, and Bossfight as environments for our main evaluation using the *easy* setting. They constitute a set of various game dynamics. While high performance in Bigfish and Starpilot is commonly achieved, Dodgeball and Bossfight are more challenging. Especially agents trained for Bossfight experience limited reward signals early in training. We provide an additional experiment that uses Coinrun and the *hard* setting; the normalization constants remain the same.

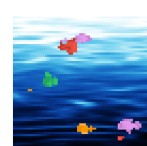 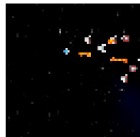 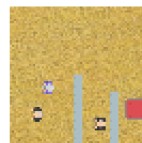 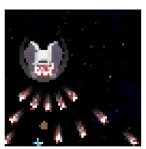 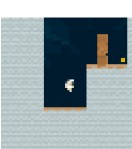

Figure A.1: Used ProcGen environments: Bigfish, Starpilot, Dodgeball, Bossfight, and Coinrun (left to right).

**Normalized Score:** As suggested by Cobbe et al. (2020), we report normalized scores $S$ by

$$S = \frac{R - R_{\min}}{R_{\max} - R_{\min}}, \tag{A.1}$$

where $R$ is the raw return collected by the agent, $R_{\min}$ is the score for the environment by a random agent, $R_{\max}$ is the maximum possible score. The normalization constants are shown in Table A.1.

Table A.1: Normalization constants for Procgen environments in the *easy* setting Cobbe et al. (2020).

| Game | $R_{\min}$ | $R_{\max}$ | Game | $R_{\min}$ | $R_{\max}$ |
|---|---|---|---|---|---|
| bigfish | 1 | 40 | jumper | 3 | 10 |
| bossfight | 0.5 | 13 | leaper | 3 | 10 |
| caveflyer | 3.5 | 12 | maze | 5 | 10 |
| chaser | 0.5 | 14 | miner | 1.5 | 14 |
| climber | 2 | 12 | ninja | 3.5 | 10 |
| coinrun | 5 | 10 | plunder | 4.5 | 30 |
| dodgeball | 1.5 | 19 | starpilot | 2.5 | 64 |
| fruitbot | -1.5 | 27 | heist | 3.5 | 10 |

# B ADDITIONAL MATERIAL FOR PRELIMINARY EXPERIMENTS

We provide additional detailed plots for our preliminary experiments on the Impala and Impoola-CNN models for PPO and DQN. Moreover, we present a supervised learning experiment to further study the effect of using the Impoola-CNN encoder.

## B.1 PRELIMINARY EXPERIMENTS FOR PPO

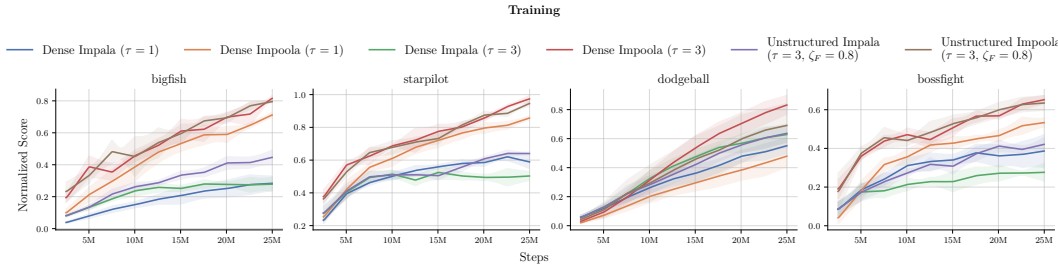

Figure B.2: Normalized scores (return) per environment of the preliminary experiments for PPO for testing levels, used to calculate the aggregated normalized scores in Figure 1.

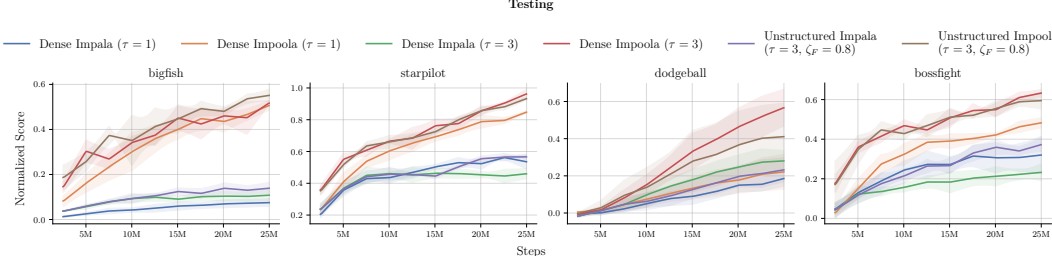

Figure B.3: Normalized scores (return) per environment of the preliminary experiments for PPO for testing levels, used to calculate the aggregated normalized scores in Figure 1.

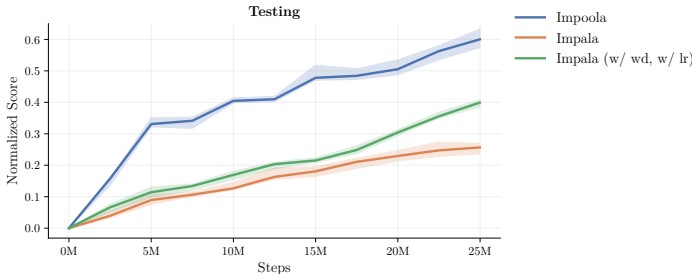

Figure B.4: Ablation for the preliminary experiments using PPO for testing levels showing that the Impoola-CNN's pooling layer is crucial as the Impala-CNN model, even when enhanced with learning rate annealing and weight decay, is not able to meet the performance of Impoola.

## B.2 PRELIMINARY EXPERIMENTS FOR DQN

Since the results by Obando-Ceron et al. (2024a) are primarily for DQN and Rainbow, we also provide experiments for DQN. Our DQN implementation uses double networks, multi-step returns, and a simplified prioritized replay buffer similar to (Obando-Ceron et al., 2024a).

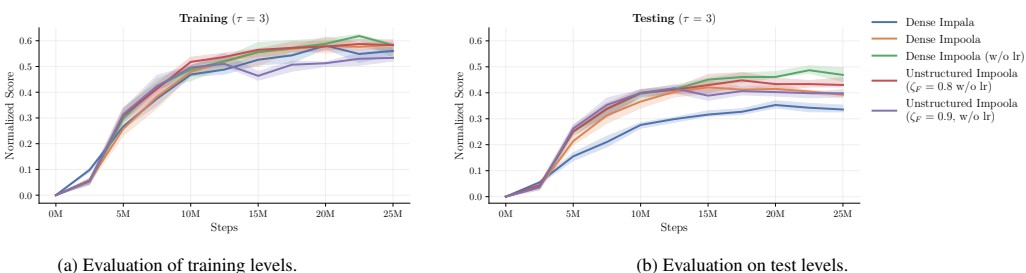

(a) Evaluation of training levels.

(b) Evaluation on test levels.

Figure B.5: Comparison of the Impala and Impoola-CNN models with scale $\tau = 3$ on the subset of four ProcGen environments for DQN training. Normalized return scores are evaluated for training levels (**left**) and test levels (**right**). Unstructured gradual pruning (Obando-Ceron et al., 2024a) results are displayed for target pruning rates of 0.8 and 0.9. The Impoola-CNN model incorporates a Pooling layer before the Flatten layer.

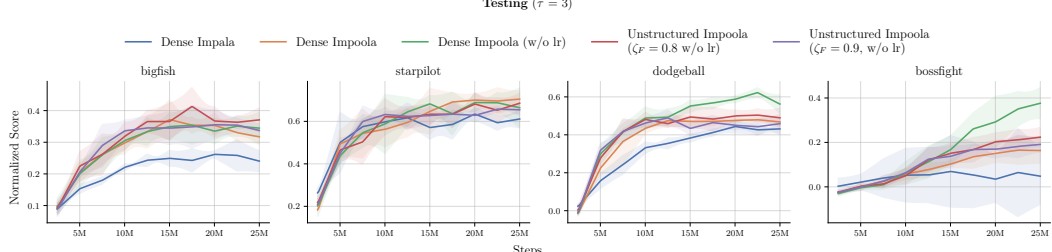

Figure B.6: Results per environment on testing levels used to calculate the normalized scores in Figure B.5.

## B.3 PRELIMINARY EXPERIMENTS FOR SUPERVISED LEARNING

We provide another experiment by using the Impoola-CNN ($\tau = 3$) for image classification of TinyImageNet as a supervised learning example. We use the exact same model architecture as for DRL with a prediction head for the 200 classes. The learning rate is set to $5e^4$ and linearly annealed. Weight decay of 1e-5 is used, but no data augmentation. Again, it can be seen in Figure B.7 that the Impoola-CNN encoder facilitates generalization while the Impala-CNN tends to overfit quickly.

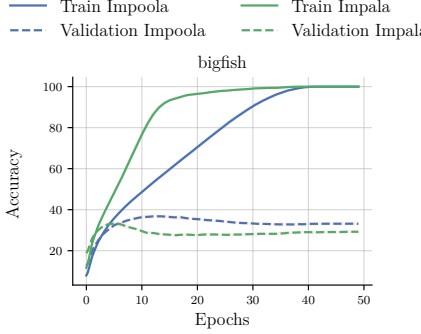

Figure B.7: Training and validation accuracy for image classification using TinyImageNet.

# C    ADDITIONAL MATERIAL FOR EXPERIMENTS

We provide additional detailed plots for our main experiments for PPO and DQN and baselines in comparison to our group-structured pruning method.

## C.1    EXPERIMENTS FOR PPO

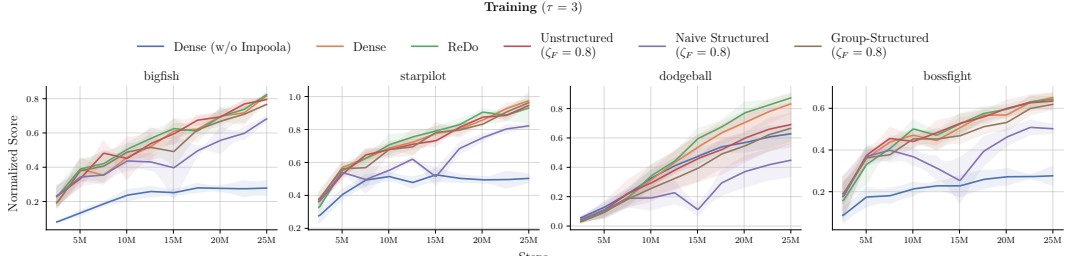

Figure C.8: Results on training levels per environment used to calculate the normalized scores in Figure 3.

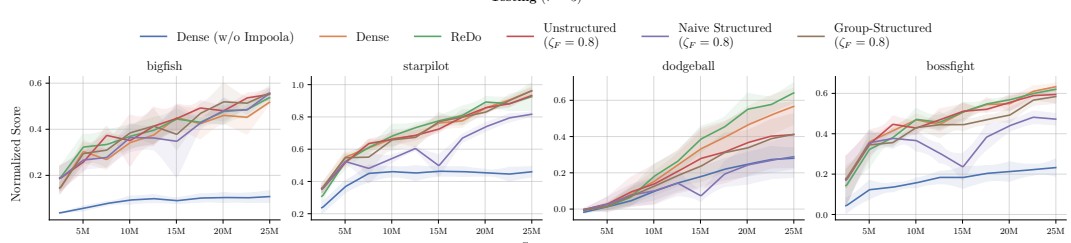

Figure C.9: Results on test levels per environment used to calculate the normalized scores in Figure 3.

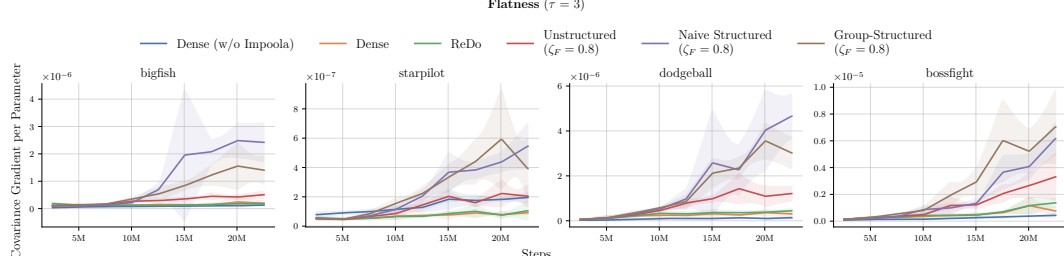

Figure C.10: Flatness of the PPO agent's DNN, measured as the gradient covariance trace per parameter during training for the generalization track. According to Bartoldson et al. (2020), flat DNNs are often associated with high generalization. We estimate flatness by the trace of the gradient's covariance matrix per parameter, with lower values suggesting better generalization. Our results reveal that pruning tends to reduce flatness. Despite this decrease in flatness, our results show that pruned networks can still generalize effectively, indicating that flatness is not the sole determinant of generalization.

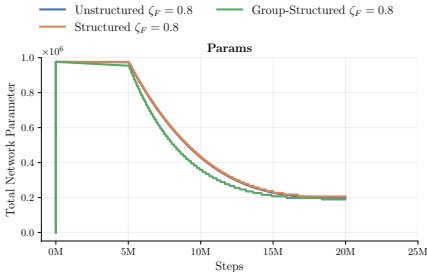

Figure C.11: Parameter counts for PPO during training for structured and unstructured pruning with a target sparsity $\zeta_f = 0.8$ and Impoola-CNN model ($\tau = 3$). It can be seen that the schedule for structured group pruning is *faster*. This is because when one structure, e.g., the input weight of a neuron, also the corresponding output weights are removed, which Equation 2 cannot account for.

## C.2 EXPERIMENTS FOR DQN

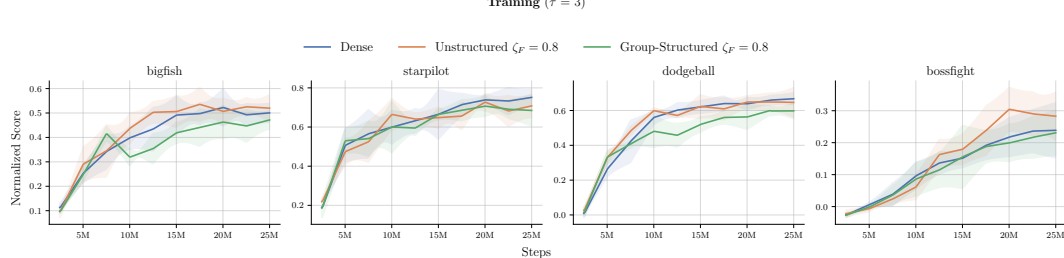

Figure C.12: Results on training levels per environment used to calculate the normalized scores in Figure 4.

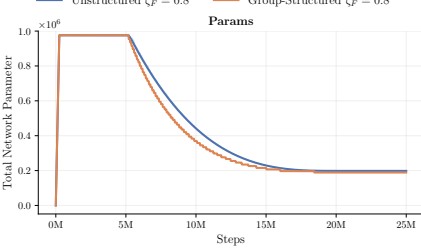

Figure C.13: Parameter counts for DQN during training for structured and unstructured pruning with a target sparsity $\zeta_f = 0.8$ and Impoola-CNN model ($\tau = 3$). It can be seen that the schedule for structured group pruning is *faster*. This is because when one structure, e.g., the input weight of a neuron, also the corresponding output weights are removed, which Equation 2 cannot account for.

# D HYPERPARAMETERS LIST

Table D.2: Hyperparameters for Proximal Policy Optimization (PPO).

| Hyperparameter | Values |
| --- | --- |
| **Number Parallel Environments** | 64 |
| **Environment Steps** | 256 |
| **Learning Rate** | $5 \times 10^{-4}$ |
| **Batch Size** | 2048 |
| **Epochs** | 3 |
| **Discount Factor** ($\gamma$) | 0.99 |
| **GAE Lambda** ($\lambda$) | 0.95 |
| **Clip Range** | 0.2 |
| **Value Function Coefficient** | 0.5 |
| **Entropy Coefficient** | 0.01 |
| **Max Gradient Norm** | 0.5 |
| **Optimizer** | Adam |
| **Shared Policy and Value Network** | Yes |

Table D.3: Hyperparameters for Deep Q-Network (DQN).

| Hyperparameter | Values |
| --- | --- |
| **Number Parallel Environments** | 64 |
| **Learning Rate** | $5 \times 10^{-5}$ |
| **Batch Size** | 256 |
| **Discount Factor** ($\gamma$) | 0.99 |
| **Target Network Update Frequency** | 64,000 steps |
| **Learning Starts** | 250,000 steps |
| **Train Frequency** | 1 |
| **Replay Buffer Size** | $1 \times 10^6$ |
| **Exploration Initial** $\epsilon$ | 1.0 |
| **Exploration Final** $\epsilon$ | 0.02 |
| **Exploration Decay Fractions** | 0.1 |
| **Max Gradient Norm** | 10.0 |
| **Optimizer** | Adam |

# E    NETWORK ARCHITECTURE

Table E.4: Model summary of the Impala network (width scale $\tau = 3$), including the Actor and Critic heads for PPO, with 64 x 64 input images. The overall parameter count is 2,450,640, with a total of 262.33M multi-adds.

| Layer (type:depth-idx) | Input | Output | Param # | Kernel | Param % | Multi-Adds |
|---|---|---|---|---|---|---|
| ImpalaPPOActorCritic | [3, 64, 64] | [15] | – | – | – | – |
| Sequential: 1-1 | [3, 64, 64] | [256] | – | – | – | – |
| ConvSequence: 2-1 | [3, 64, 64] | [48, 32, 32] | – | – | – | – |
| Conv2d: 3-1 | [3, 64, 64] | [48, 64, 64] | 1,344 | [3, 3] | 0.05% | 5,505,024 |
| ResidualBlock: 3-2 | [48, 32, 32] | [48, 32, 32] | – | – | – | – |
| Conv2d: 4-1 | [48, 32, 32] | [48, 32, 32] | 20,784 | [3, 3] | 0.85% | 21,282,816 |
| Conv2d: 4-2 | [48, 32, 32] | [48, 32, 32] | 20,784 | [3, 3] | 0.85% | 21,282,816 |
| ResidualBlock: 3-3 | [48, 32, 32] | [48, 32, 32] | – | – | – | – |
| Conv2d: 4-3 | [48, 32, 32] | [48, 32, 32] | 20,784 | [3, 3] | 0.85% | 21,282,816 |
| Conv2d: 4-4 | [48, 32, 32] | [48, 32, 32] | 20,784 | [3, 3] | 0.85% | 21,282,816 |
| ConvSequence: 2-2 | [48, 32, 32] | [96, 16, 16] | – | – | – | – |
| Conv2d: 3-4 | [48, 32, 32] | [96, 32, 32] | 41,568 | [3, 3] | 1.70% | 42,565,632 |
| ResidualBlock: 3-5 | [96, 16, 16] | [96, 16, 16] | – | – | – | – |
| Conv2d: 4-5 | [96, 16, 16] | [96, 16, 16] | 83,040 | [3, 3] | 3.39% | 21,258,240 |
| Conv2d: 4-6 | [96, 16, 16] | [96, 16, 16] | 83,040 | [3, 3] | 3.39% | 21,258,240 |
| ResidualBlock: 3-6 | [96, 16, 16] | [96, 16, 16] | – | – | – | – |
| Conv2d: 4-7 | [96, 16, 16] | [96, 16, 16] | 83,040 | [3, 3] | 3.39% | 21,258,240 |
| Conv2d: 4-8 | [96, 16, 16] | [96, 16, 16] | 83,040 | [3, 3] | 3.39% | 21,258,240 |
| ConvSequence: 2-3 | [96, 16, 16] | [96, 8, 8] | – | – | – | – |
| Conv2d: 3-7 | [96, 16, 16] | [96, 16, 16] | 83,040 | [3, 3] | 3.39% | 21,258,240 |
| ResidualBlock: 3-8 | [96, 8, 8] | [96, 8, 8] | – | – | – | – |
| Conv2d: 4-9 | [96, 8, 8] | [96, 8, 8] | 83,040 | [3, 3] | 3.39% | 5,314,560 |
| Conv2d: 4-10 | [96, 8, 8] | [96, 8, 8] | 83,040 | [3, 3] | 3.39% | 5,314,560 |
| ResidualBlock: 3-9 | [96, 8, 8] | [96, 8, 8] | – | – | – | – |
| Conv2d: 4-11 | [96, 8, 8] | [96, 8, 8] | 83,040 | [3, 3] | 3.39% | 5,314,560 |
| Conv2d: 4-12 | [96, 8, 8] | [96, 8, 8] | 83,040 | [3, 3] | 3.39% | 5,314,560 |
| Flatten: 2-4 | [96, 8, 8] | [6144] | – | – | – | – |
| ReLU: 2-5 | [6144] | [6144] | – | – | – | – |
| Linear: 2-6 | [6144] | [256] | 1,573,120 | – | 64.19% | 1,573,120 |
| ReLU: 2-7 | [256] | [256] | – | – | – | – |
| Linear: 1-2 | [256] | [15] | 3,855 | – | 0.16% | 3,855 |
| Linear: 1-3 | [256] | [1] | 257 | – | 0.01% | 257 |

Table E.5: Model summary of the Impoola network (width scale $\tau = 3$), including the Actor and Critic heads for PPO, with 64 x 64 input images. The overall parameter count is 976,080, with a total of 260.85M multi-adds.

| Layer (type:depth-idx) | Input | Output | Param # | Kernel | Param % | Multi-Adds |
|---|---|---|---|---|---|---|
| ImpalaPPOActorCritic | [3, 64, 64] | [15] | – | – | – | – |
| Sequential: 1-1 | [3, 64, 64] | [256] | – | – | – | – |
| ConvSequence: 2-1 | [3, 64, 64] | [48, 32, 32] | – | – | – | – |
| Conv2d: 3-1 | [3, 64, 64] | [48, 64, 64] | 1,344 | [3, 3] | 0.14% | 5,505,024 |
| ResidualBlock: 3-2 | [48, 32, 32] | [48, 32, 32] | – | – | – | – |
| Conv2d: 4-1 | [48, 32, 32] | [48, 32, 32] | 20,784 | [3, 3] | 2.13% | 21,282,816 |
| Conv2d: 4-2 | [48, 32, 32] | [48, 32, 32] | 20,784 | [3, 3] | 2.13% | 21,282,816 |
| ResidualBlock: 3-3 | [48, 32, 32] | [48, 32, 32] | – | – | – | – |
| Conv2d: 4-3 | [48, 32, 32] | [48, 32, 32] | 20,784 | [3, 3] | 2.13% | 21,282,816 |
| Conv2d: 4-4 | [48, 32, 32] | [48, 32, 32] | 20,784 | [3, 3] | 2.13% | 21,282,816 |
| ConvSequence: 2-2 | [48, 32, 32] | [96, 16, 16] | – | – | – | – |
| Conv2d: 3-4 | [48, 32, 32] | [96, 32, 32] | 41,568 | [3, 3] | 4.26% | 42,565,632 |
| ResidualBlock: 3-5 | [96, 16, 16] | [96, 16, 16] | – | – | – | – |
| Conv2d: 4-5 | [96, 16, 16] | [96, 16, 16] | 83,040 | [3, 3] | 8.51% | 21,258,240 |
| Conv2d: 4-6 | [96, 16, 16] | [96, 16, 16] | 83,040 | [3, 3] | 8.51% | 21,258,240 |
| ResidualBlock: 3-6 | [96, 16, 16] | [96, 16, 16] | – | – | – | – |
| Conv2d: 4-7 | [96, 16, 16] | [96, 16, 16] | 83,040 | [3, 3] | 8.51% | 21,258,240 |
| Conv2d: 4-8 | [96, 16, 16] | [96, 16, 16] | 83,040 | [3, 3] | 8.51% | 21,258,240 |
| ConvSequence: 2-3 | [96, 16, 16] | [96, 8, 8] | – | – | – | – |
| Conv2d: 3-7 | [96, 16, 16] | [96, 16, 16] | 83,040 | [3, 3] | 8.51% | 21,258,240 |
| ResidualBlock: 3-8 | [96, 8, 8] | [96, 8, 8] | – | – | – | – |
| Conv2d: 4-9 | [96, 8, 8] | [96, 8, 8] | 83,040 | [3, 3] | 8.51% | 5,314,560 |
| Conv2d: 4-10 | [96, 8, 8] | [96, 8, 8] | 83,040 | [3, 3] | 8.51% | 5,314,560 |
| ResidualBlock: 3-9 | [96, 8, 8] | [96, 8, 8] | – | – | – | – |
| Conv2d: 4-11 | [96, 8, 8] | [96, 8, 8] | 83,040 | [3, 3] | 8.51% | 5,314,560 |
| Conv2d: 4-12 | [96, 8, 8] | [96, 8, 8] | 83,040 | [3, 3] | 8.51% | 5,314,560 |
| AdaptiveAvgPool2d: 2-4 | [96, 8, 8] | [96, 2, 2] | – | – | – | – |
| Flatten: 2-5 | [96, 2, 2] | [384] | – | – | – | – |
| ReLU: 2-6 | [384] | [384] | – | – | – | – |
| Linear: 2-7 | [384] | [256] | 98,560 | – | 10.10% | 98,560 |
| ReLU: 2-8 | [256] | [256] | – | – | – | – |
| Linear: 1-2 | [256] | [15] | 3,855 | – | 0.39% | 3,855 |
| Linear: 1-3 | [256] | [1] | 257 | – | 0.03% | 257 |

## F  MEASURED INFERENCE TIMES

We measure the results after a warm-up phase of 100 forward passes as the average of 1000 forward passes without further soft- or hardware optimizations.

The used devices are :

- **High-end GPU:** NVIDIA RTX A6000
- **Workstation CPU:** Intel Xeon W-2295
- **Embedded CPU**: NVIDIA Jetson Orin Nano 7W
- **Embedded GPU:** NVIDIA Jetson Orin Nano 7W.

Table F.6: Comparison of latency times in ms across compute devices for the presented pruning algorithms using the Impoola-CNN model with a width scale of $\tau = 5$.

| Compute Device | Batch Size 256 | | | | | Batch Size 1 | | | | |
| --- | --- | --- | --- | --- | --- | --- | --- | --- | --- | --- |
| | | Unstructured | | Group-Structured | | | Unstructured | | Group-Structured | |
| | Dense | $\zeta_F$=0.8 | $\zeta_F$=0.9 | $\zeta_F$=0.8 | $\zeta_F$=0.9 | Dense | $\zeta_F$=0.8 | $\zeta_F$=0.9 | $\zeta_F$=0.8 | $\zeta_F$=0.9 |
| High-end GPU | 21.1 | 20.7 | - | 10.1 | - | 1.0 | 1.0 | - | 1.0 | - |
| Workstation CPU | 637.6 | 619.9 | - | 251.6 | - | 4.0 | 4.1 | - | 2.7 | - |
| Embedded GPU | 865.8 | 865.8 | - | 506.1 | - | 8.8 | 8.8 | - | 7.1 | - |
| Embedded CPU | - | - | - | - | - | 54.1 | 54.9 | - | 23.2 | - |

