# OpenReview forum: "The Case for Gradual Structured Pruning in Image-based Deep Reinforcement Learning"
_ICLR.cc/2025/Conference — Submitted to ICLR 2025_

### Official Review · Reviewer_Dak6 · 2024-10-27

**Soundness:** 2
**Presentation:** 2
**Contribution:** 2
**Rating:** 5
**Confidence:** 4

**Summary:**

Scaling neural networks has generally been ineffective for deep reinforcement learning, as larger networks do not necessarily improve performance. Recent work by Obando-Ceron et al. (2024) suggests that pruning a large network could overcome this limitation. By initially increasing the network size and applying pruning during training, they achieved performance gains beyond those possible with a standard dense network. However, their unstructured sparsity approach often fails to generalize well outside the original setting. This paper introduces an improved CNN architecture, called Impoola, along with a structured pruning algorithm that outperforms unstructured pruning in both generalization and computational efficiency.

**Strengths:**

- Exploring scaling laws for deep RL is an important challenge.
- New CNN architecture and structured pruning algorithm are proposed.
- Outperform both unstructured and naive structured pruning baselines.

**Weaknesses:**

**Clarification of objectives and contribution**

The primary goal of this paper—whether to build upon Obando-Ceron et al.'s (2024) approach for finding scalable architectures or to develop a structured pruning method specifically for deep RL—is somewhat unclear. If the objective is scaling, adding experiments that compare the proposed method with the original dense network and unstructured pruning across different scenarios would strengthen its claims. On the other hand, if structured pruning is the focus, revisiting the introduction and motivation to reflect this intent could help clarify the paper’s contributions in this area.

---
**Need for broader benchmark comparisons**

If this paper aims to address scaling issues in deep RL, it would benefit from broader comparisons with relevant methods, such as those presented in [1] and [2]. Including at least one of these as a baseline would provide a more comprehensive context for the paper’s approach. If these methods are excluded, providing a brief rationale would clarify the baseline selection choices for readers.

---
**Potential for alternative focus on structured pruning**

An alternative approach for the paper could be to frame it as a structured pruning study specifically for deep RL, rather than as an extension of scaling work. The proposed Impoola architecture appears specialized, so comparing it with existing structured pruning methods for deep RL, such as [3], would reinforce its relevance in this area. Additionally, if the technique is intended to be broadly applicable to CNNs, testing it on traditional CNN tasks (e.g., video classification) would better demonstrate its versatility. Including comparisons with CNN-focused pruning techniques, such as [4], would further highlight the method’s adaptability across different applications.



[1] Obando-Ceron et al. Mixtures of Experts Unlock Parameter Scaling for Deep RL. ICML 2024.\
[2] Sokar et al. Don't Flatten, Tokenize! Unlocking the Key to SoftMoE's Efficacy in Deep RL. arXiv 2024.\
[3] Su et al. Compressing Deep Reinforcement Learning Networks with a Dynamic Structured Pruning Method for Autonomous Driving. arXiv 2024.\
[4] He & Xiao. Structured Pruning for Deep Convolutional Neural Networks: A Survey. TPAMI 2023.

**Questions:**

N/A

---

> ### Author Response · Authors · 2024-11-28
> **Answer to Questions**
>
> Dear Reviewer,
>
> We tried to address your specific concerns as follows:
>
> ### Clarification of Objective:
> We see the need to clarify the objective of our work more clearly, which is twofold: it is about scaling but in an “efficient” way, thus the structured pruning. We revised large parts of the introduction to discuss the scaling aspect first; please see the new Figure 1. Our main finding is that when using Impoola, we can actually unlock performance gains by scaling without the need for unstructured pruning. However, this brings us to the aspect of increased memory size and inference times. When DRL is deployed for robotics, this typically happens on resource-limited embedded boards, so this aspect matters a lot. We address this aspect with our group-structured pruning method, showing that performance comparable to the dense baselines can be achieved while improving inference times (and even training time). We added an ablation to Section 5.4 for Impala to show that group-structured pruning also works for this case; however, overall performance is inferior to that of the Impoola case.
>
> ### Broader Benchmark Comparision
> We tried to address your requested benchmark comparison by introducing a policy distillation approach with a smaller student network, see Table 1. The methods presented in [1, 2] only focus on the scaling aspect, and our objective is to scale efficiently without an increase in computation requirements, which can be seen as a different direction. Thus, the distillation method is a closer method as it evaluates how well performance from a large dense model can be distilled into a smaller one. We show that gradual pruning is way more effective in achieving performance gains and limiting inference time increases. We also highlight the reduced training time when using group-structured pruning in Table 1.
>
> ### Alternative Focus
> The motivation for our work is two-fold: scaling and pruning but in an efficient way, so our study is about the combination. As the application of our work is for DRL, we focus on results there. However, we did see the need for further results to understand and prove the broader applicability, so we conducted further experiments for Atari games (see Section 5.4). Our results show the usefulness of Impoola and group-structured pruning for Atari, proving their wide applicability in image-based DRL (which is the intended use).
>
> The work of Su et al. [3] cannot really be compared to our method as it is not designed for image-based DRL and uses only linear layers. While they use the formulation of a group, it simply considers the input and output weights of a linear layer’s neurons, while our ResNet-based architecture has complex intra- and interlayer dependencies. However, we still decided to list their work in the related work section. Further (structured) pruning methods from [4] will be interesting future work but are out of scope. Our work can be seen as an initial study for structured pruning of scaled networks in image-based DRL.

---

> > ### Comment · Reviewer_Dak6 · 2024-12-01
> > **Response to the Rebuttal**
> >
> > Thank you for the rebuttal. The revised emphasis on "efficient scaling" is clearer and more reasonable. However, the overall presentation does not fully support this message. For instance, Figure 1 does not effectively demonstrate the scaling benefits of structured pruning. Similarly, the main results in Table 1 do not align with the message, as the dense configuration achieves the best performance.
> >
> > To address this, the main results should clearly show:
> > 1. Both structured and unstructured pruning enable scaling, unlike dense networks.
> > 2. Structured pruning outperforms unstructured pruning for the same computational budget.
> >
> > While I have raised my rating, I still recommend a weak rejection due to these concerns. Reorganizing the flow and results to better support these points would greatly strengthen the overall message.

---

> > > ### Author Response · Authors · 2024-12-02
> > > **Clarification about Scaling**
> > >
> > > Dear Reviewer,
> > >
> > > We appreciate your effort in explaining your decision in detail, that’s very helpful. We would like to clarify a crucial aspect related to your first point, as this is the main motivation for our work.
> > >
> > > If we use the Impoola-CNN, we can unlock performance gains through network scaling **without** the need for any pruning. This is the key message of Figure 1; we discuss and highlight this several times in the manuscript, e.g., lines 48-50: “We show in the following preliminary experiment that the benefits of network scaling in image-based DRL can also be unlocked by simple architectural refinements of the Impala-CNN, rendering the use of unstructured pruning from Obando-Ceron et al. (2024a) for performance increase obsolete.” This means that for Impoola, dense networks seem to be the upper ceiling in performance, and the motivation for pruning should be to reduce computation requirements. We reason why pruning does not further increase performance for Impoola in the discussion section (lines 536-539): "The fact that pruning methods seem to not outperform the Dense networks for the Impoola-CNN... can be related to the finding of Cobbe et al., 2019 that if some regularization is already applied, the combination with further regularization technique does not necessarily improve performance further." This finding motivates us then to propose our (group-) structured pruning to maintain computational efficiency when scaling the network, as for Impoola, network scaling itself already improves performance.
> > >
> > > Figure 1 is a preliminary experiment that only shows that pruning is not necessary for performance through network scaling since Impoola allows for that. Thus, (group-) structured pruning results are only shown in our main experiments. Note that there is also an ablation comparing Impoola and Impala in Figure 9.
> > >
> > > Figure 7 actually contains your requested comparison, but unfortunately, we did not discuss it that way. Group-structured pruning with scale 3 has a comparable budget to dense/unstructured pruning with scale 1, and the figure shows that group-structured with scale 3 clearly outperforms networks with scale 1. We plan to make this comparison more explicit in the future as it serves our work's motivation well.

---

> ### Comment · Reviewer_Dak6 · 2024-12-02
> **Response to the Rebuttal**
>
> Thank you for the kind explanation. I understand that Figure 1 presents the preliminary experiments and Figure 7 contains some relevant information. However, the current presentation and results do not clearly convey the revised message of efficient scaling. Reorganizing the overall flow and aligning each figure and table consistently with the message would greatly enhance the paper's clarity and impact.

---

### Official Review · Reviewer_a9a9 · 2024-11-01

**Soundness:** 3
**Presentation:** 2
**Contribution:** 3
**Rating:** 6
**Confidence:** 3

**Summary:**

This paper proposes a gradual structured pruning method for deep reinforcement learning. The authors firstly analyzes the drawbacks of the unstructured pruning, then introduces a improved impoola-CNN model to replace the previous impala-CNN model, and a gradual structured pruning for image-based deep reinforcement learning tasks. The experimental results show that the proposed method has compariable perfermances comparing with unstrutured pruning mehtod, and it has good robustness when intruducing nosies. Moreover, the proposed pruning method results in less latency times comparing with original dense model and unstructured pruned model.

**Strengths:**

1. The gradual structured pruning method is well-defined, and the experimental results show that is has good computation efficiency

**Weaknesses:**

1. The experimental results did not include the comparsion of impala-CNN and impoola-CNN, thus the advantages of the new network cannot be proved. The authors may consider to provide a direct comparison of performance between Impala-CNN and Impoola-CNN under the same pruning conditions across multiple environments, as well as the un-pruned editions. This would give clearer evidence for the claimed advantages of the new architecture.

2. The advantages of the scoring function are not proved. The reason that use ||w||_1 should be carefully analyzed. For instance, the authors may consider to compare L1-norm, L2-norm or other regularization methods to show the advantages of the proposed scoring function.

**Questions:**

1. Is their some comparsion between impala-CNN and impoola-CNN when both of them are processed by using gradual structured pruning? The authors may use the proposed pruning mehtod to both impala-CNN and impoola-CNN under different configurations to show the improvements introduced by impoola-CNN.

2. Is their any experimental results to compare the performance of using different scoring functions?

---

> ### Author Response · Authors · 2024-11-28
> **Answer to Questions**
>
> Dear Reviewer,
>
> We tried to address your specific concerns as follows:
>
> ### Questions 1:
> We conducted additional experiments to compare Impoola and Impala using our group-structured pruning and included them in Section 5.4 Ablations. It can be seen that overall, Impoola, with any pruning methods, is superior to Impala. The new results for Impala show that group-structured pruning also improves performance compared to dense Impala; however, the gains are less than for unstructured pruning (group-structured pruning seems to work better when the training is stable, e.g., when using Impoola).
>
> This question also relates well to our reformulated introduction with a hopefully more explicit motivation. We replaced Figure 1 to compare performance for network width scales of $\tau=1$ and $\tau=3$ for Impala and Impoola. The main finding can be summarized as follows: Impoola allows for scaling (due to the pooling layer), and unstructured pruning is not needed as for Impala. This motivates us to find a way, namely our group-structured method, to reduce compute times as this will grow for scaled networks.
>
> ### Question 2:
> We used $||w||_1$ as a scoring function, which was proven to work well in other works [1]. We provide a new ablation on the scoring function in Section 5.4 Ablations for L1, L2, Taylor expansion, and random selection. There might be a slight performance increase when using the Taylor expansion. However, L1 should be preferred due to its simplicity. Interestingly, random selection also performs quite well, which aligns with other works on random pruning [2].
>
> [1] Johan Samir Obando-Ceron, Aaron Courville, and Pablo Samuel Castro. In value-based deep reinforcement learning, a pruned network is a good network. In Forty-first International Conference on Machine Learning, 2024
>
> [2] Shiwei Liu, Tianlong Chen, Xiaohan Chen, Li Shen, Decebal Constantin Mocanu, Zhangyang Wang, and Mykola Pechenizkiy. The unreasonable effectiveness of random pruning: Return of the most naive baseline for sparse training. In International Conference on Learning Representations, 2022.

---

> > ### Comment · Reviewer_a9a9 · 2024-11-29
> >
> > Thank you for your rebuttal. I believe that your responses can solve my concerns, thus I would like to update my score to 6.

---

### Official Review · Reviewer_Xmu9 · 2024-11-01

**Soundness:** 2
**Presentation:** 3
**Contribution:** 2
**Rating:** 3
**Confidence:** 4

**Summary:**

The paper introduces a group-structured pruning framework for image-based deep reinforcement learning (DRL), specifically designed to maintain performance while reducing computational costs. The authors first demonstrate that the benefits of unstructured pruning diminish when making architectural improvements to neural networks. They then propose a structured pruning approach that removes entire network structures (channels/neurons) while accounting for inter-layer dependencies. The method is extensively evaluated on the Procgen Benchmark using PPO and DQN agents, showing comparable performance to unstructured pruning while achieving significant reductions in inference time.

**Strengths:**

1. The overall writing is clear.
2. Efficiency in the domain of RL is interesting.

**Weaknesses:**

1. The motivation is not clear. Although efficiency in the domain of RL is interesting, the author should include the difference between "pruning in image classification" and "pruning in reinforcement learning." This motivation should serve the algorithm.

2. There are some other references [a] to do gradual pruning. What is the difference? Why not cite this?

3. Some arguments are wrong. For example, in line 15, the authors claim, "unstructured pruning merely zeroes out individual weights, the resulting networks usually retain high computational demands despite sparsity". This is not correct. For example, Thinet [b] can achieve realistic acceleration. There are so many other filter pruning methods to achieve practical acceleration.

4. The performance is not good. In table 1, if the up arrow means "higher is better", the proposed method is not as good as others.

[a] H. Wang, C. Qin, Y. Zhang, and Y. Fu, “Neural pruning via growing regularization,” in Proc. Int. Conf. Learn. Represent., 2022
[b] https://github.com/Roll920/ThiNet

**Questions:**

See weaknesses

---

> ### Author Response · Authors · 2024-11-21
> **Clarification about Weakness 3**
>
> Dear Reviewer,
>
> Thank you for your helpful feedback!
>
> We are currently revising our manuscript, motivating our work more clearly as a method for efficient scaling of DNNs in image-based DRL. We will upload a revision and full answers soon but would like to ask for an initial clarification about Weakness 3:
>
> From reading the Thinet paper of Luo et al. [b] and looking at their implementation, we understand their method as structured pruning, just as we do. They remove the filters corresponding to an output channel, which has little importance for the next layer. This is what we call “structured” pruning of channels, or more precisely, the filter(s) creating that output channel. Luo et al. also mention that speed-ups cannot be measured for **un**structured pruning, where weights without any pattern are removed/set to zero.
>
> Taken from Luo et al.: “... non-structured sparse model can not be supported by off-the-shelf libraries, thus specialized hardwares and softwares are needed for efficient inference, which is difficult and expensive in real-world applications. On the other hand, the non-structured random connectivity ignores cache and memory access issues. As indicated in [32], due to the poor cache locality and jumping memory access caused by random connectivity, the practical acceleration is very limited (sometimes even slows down), even though the actual sparsity is relatively high.”
>
> Therefore, we interpret Luo et al. [b] as supporting our argument about **un**structured pruning and that only structured pruning has the advantage of measurable speed-ups.
>
> Could you please let us know if this clarification addresses your doubts about our argument?
>
> [b] Luo, Jian-Hao, Jianxin Wu, and Weiyao Lin. "Thinet: A filter level pruning method for deep neural network compression." Proceedings of the IEEE international conference on computer vision. 2017.

---

> > ### Comment · Reviewer_Xmu9 · 2024-11-24
> >
> > Thanks for your response.
> >
> > I'm still waiting for your response to other reviewers' comments.

---

> ### Author Response · Authors · 2024-11-28
> **Answer to Questions**
>
> Dear Reviewer,
>
> We tried to address your specific concerns as follows:
>
> ### Weakness 1:
> We decided to reformulate the introduction as the motivation for our work is actually two-fold: Scaling and pruning in combination for efficient scaling. We replaced Fig. 1 to compare the performance for network width scales of $\tau=1$ and $\tau=3$. This preliminary result concludes that incorporating a pooling layer in the Impoola-CNN allows us to realize performance gains when scaling the network, which is not the case for Impala. While we can confirm the claims of [1] that pruning increases performance when scaling Impala, we demonstrate that Impoola performs significantly better overall. As it can be seen that pruning is slightly detrimental to the performance of Impoola, the use of **un**structured pruning for performance increase becomes obsolete. However, we use this finding to motivate the use of (group-) structured pruning instead to unlock the performance gains from scaled Impoola in combination with lowering the compute demand again for scaled networks. We think this has a high practical appeal for real-world DRL applications. As Impala/Impoola has inter-layer dependencies due to the residual connections, we present a group-structured pruning method that considers these dependencies.
>
> We added the following to discuss the differences between pruning in image classification and DRL further: "Network pruning is a widely used technique in other deep learning fields, e.g., image classification (Vadera & Ameen, 2022), originally aimed at reducing DNNs’ memory footprint and inference time but also known to frequently enhance robustness and generalization (Bartoldson et al., 2020). Its use in DRL may introduce advantageous regularization (Obando-Ceron et al., 2024a) but poses a unique challenge due to its dynamic training, requiring methods that maintain training stability over time.”
>
> ### Weakness 2:
> We updated the relevant Sections 2. and 3.2 to emphasize that gradual pruning in DRL
> is a common practice for DRL, introduced by [2] and used in [1]. As this is no novelty from our side, we referred to the survey of [3] for further works in other fields, as the gradual pruning itself is no focus of our work. We hope that the new version emphasizes this fact more now.
>
> ### Weakness 3:
> We hope that our earlier reply addressed this point adequately. Please let us know if you still have concerns.
>
> ### Weakness 4:
> Our revision highlights the advantages of our method more clearly and puts the performance into a better perspective. As we discussed, dense Impoola performs the best regarding reward maximization and can be understood as the upper ceiling (it outperforms Impala and pruning). Our goal is now to meet this performance as closely as possible with pruned versions, where our group-structured pruning method performs as well as unstructured pruning but with a significant reduction of inference and training time (added to the table). Thus, group structure enables low inference times in combination with performance gains from network scaling. We added a distillation method as an additional comparison to our results table to show that pruning during training is better than static offline distillation. We also added further experiments to emphasize the performance improvements of Impoola vs Impala and results for Atari games to highlight the broad applicability.
>
> [1] Johan Samir Obando-Ceron, Aaron Courville, and Pablo Samuel Castro. In value-based deep reinforcement learning, a pruned network is a good network. In Forty-first International Conference on Machine Learning, 2024
>
> [2] Laura Graesser, Utku Evci, Erich Elsen, and Pablo Samuel Castro. The state of sparse training in deep reinforcement learning. In International Conference on Machine Learning, pp. 7766–7792. PMLR, 2022.
>
> [3] Hongrong Cheng, Miao Zhang, and Javen Qinfeng Shi. A survey on deep neural network pruning: Taxonomy, comparison, analysis, and recommendations. IEEE Transactions on Pattern Analysis and Machine Intelligence, 2024.
>
> [4] Jian-Hao Luo, Jianxin Wu, and Weiyao Lin. Thinet: A filter level pruning method for deep neural network compression. In Proceedings of the IEEE international conference on computer vision, pp. 5058–5066, 2017.

---

> > ### Comment · Reviewer_Xmu9 · 2024-12-02
> >
> > Thanks for the effort.
> > After reading the rebuttal, I believe it's still like a combination of RL and pruning. The uniqueness of the combination is not convincing to me. I would like to keep the rating.

---

> ### Author Response · Authors · 2024-12-02
>
> Dear Reviewer,
>
> Thanks for reading the revision!
>
> We just want to give a quick follow-up to make sure there is no misconception. First, we would like to confirm that, just as you believe, the work is about combining pruning and DRL **when scaling networks**. Second, the uniqueness is given as we present a method that maintains computational efficiency when networks are scaled in DRL, a situation of high practical relevance. We think efficiency is an important topic in DRL, especially for robotic applications and deployment on embedded devices.
>
> Please let us know if you would like further clarification on these points.

---

### Author Response · Authors · 2024-11-28
**List of Changes**

Dear Reviewers,

Thank you for your constructive feedback. We uploaded the revision of the paper and hope the reviewers have the time to check the updated manuscript to see whether their concerns were sufficiently addressed.

**List of overall major changes (chronologically)**:
- Reformulation of the motivation. The preliminary experiment and Figure 1 were updated to emphasize that using Impoola unlocks performance gains through network scaling, but without the need for **un**structured pruning. This serves as motivation for our algorithm, which unlocks performance gains from scaling in combination with lowering compute demand substantially for such scaled networks.
- Extending Section 2. Related Work with a paragraph on Scaling in DRL.
- Table 1 is extended by results for Impala (Dense w/o Impoola) and we added another baseline method (Distillation BC) for better comparison. We also included the training time of each method. This should highlight the performance of group-structured pruning better, as the aim is to develop a method that combines performance improvement (Impoola) and lowering the compute demand of such scaled networks (group-structured pruning).
- Section 5.4 Ablations included several new experiments. There is a new ablation on the importance score function (L1, L2, Taylor, random). We provide further results for a comparison of pruned Impoola and Impala networks.
- We conducted additional experiments of our method for Atari games, confirming the results for Procgen. This demonstrates the broad applicability of our approach.
- **Important note**: We updated the paper's title to reflect our focus on efficient scaling: Gradual Structured Pruning for Efficient Network Scaling in Image-Based Deep Reinforcement Learning.

---

### Meta-Review · Area_Chair_jt65 · 2024-12-23

**Metareview:**

This submission proposes a gradual structured pruning method for deep reinforcement learning. After analyzing the drawbacks of prior work and unstructured pruning, they propose an improved Impoola-CNN model and demonstrate comparable performance to unstructured pruning.
The submission received ratings of 3, 5, 6.
Multiple reviewers highlighted issues with the organization of the paper, making its key goals and objectives unclear.
Even after the rebuttal, the reviewers highlighted the issues during the AC discussion. After reading through the submission, the ACs concur with the reviewers regarding it needing significant changes to reach the bar for acceptance.
The ACs did not find enough reason to overturn the overall negative consensus and recommend rejection.

**Additional Comments On Reviewer Discussion:**

During the discussion, reviewers Xmu9 and Dak6 reiterated that the major contribution of this submission was still not clear and recommended further revisions to improve its quality. They both recommended rejection. After reading through the submission, the ACs agree with the reviewers' assessments and do not find enough reason to overturn the consensus.

---

### Decision · Program_Chairs · 2025-01-22

Reject